# Bamboo Shoots Modulate Gut Microbiota, Eliminate Obesity in High-Fat-Diet-Fed Mice and Improve Lipid Metabolism

**DOI:** 10.3390/foods12071380

**Published:** 2023-03-24

**Authors:** Xiaolu Zhou, SolJu Pak, Daotong Li, Li Dong, Fang Chen, Xiaosong Hu, Lingjun Ma

**Affiliations:** National Engineering Research Center for Fruit and Vegetable Processing, Key Laboratory of Fruit and Vegetables Processing Ministry of Agriculture, College of Food Science and Nutritional Engineering, China Agricultural University, Beijing 100083, China; zxl782072778@163.com (X.Z.); ogj1991928@163.com (S.P.); lidaotong@bjmu.edu.cn (D.L.); li_dong127@163.com (L.D.); chenfangch@sina.com (F.C.); huxiaos@263.net (X.H.)

**Keywords:** bamboo shoots, fecal metabolomics, gut microbiota, insulin resistance, obesity

## Abstract

Bamboo shoots (BS) have a variety of nutritional benefits; however, their anti-obesity effect and its underlying mechanism of action are still unclear. In this study, we investigated the protective effect of BS against high-fat diet (HFD)-induced gut dysbiosis in mice. After 12 weeks of feeding C57BL/6J mice either on a normal or an HFD with or without BS, metabolic indicators, including blood lipids and glucose tolerance, were measured. *16S rRNA* gene sequencing and metabolomics were used to identify alterations in gut microbiota composition and fecal metabolic profiling. The results demonstrated that BS supplementation reduced body weight by 30.56%, mitigated liver damage, and improved insulin resistance and inflammation in obese mice. In addition, BS increased short-chain fatty acid (SCFA) levels and SCFA-producing bacteria (e.g., *Lachnospiraceae_NK4A136_group* and *Norank_f_Muribaculaceae*), and reduced levels of harmful bacteria (e.g., *Blautia* and *Burkholderia–Paraburkholderia*). Finally, BS increased many beneficial fecal metabolites, such as fatty acids and bile acids, which are highly relevant to the altered gut microbiota. Based on the modulatory effect of BS on microbiota composition and gut metabolite levels observed in this study, we suggest that BS may be beneficial in treating obesity and its related complications.

## 1. Introduction

Obesity can induce a variety of chronic metabolic diseases and disorders, such as insulin resistance and type 2 diabetes mellitus (T2DM) [1]. Obesity is a multifactorial disease, and its pathogenesis involves multiple genetic, socio-economic, and cultural factors. There are complex interactions among these factors which result in excessive fat accumulation [2]. Fat accumulation is mainly achieved by increasing the number and the size of cells available to store energy in white fat [3]. When fat cells increase in size to a certain point, the cells are stressed, and this stress induces an inflammatory response in the fat and liver tissues. The tissue inflammatory response, in turn, activates immune cells to produce more pro-inflammatory factors, exacerbating the inflammatory response [4]. At the same time, excessive accumulation of fat can lead to ectopic fat deposits in the liver and other tissues [5]. To summarize, obesity negatively impacts the host’s overall metabolism and health [6,7,8].

The gut microbiota is closely related to the occurrence and development of obesity due to its participation in nutrient absorption, energy production, and fat storage [1]. In one study, germ-free mice did not become obese even after eating a high-fat diet, suggesting that gut microbiota may have a causal role in obesity [9]. Gut microbiota may affect obesity directly, or it may play an indirect role through gut microbiota metabolites [10]. For example, the short-chain fatty acids (SCFAs) produced by the gut microbiota through the bacterial metabolism of carbohydrates can regulate glucose homeostasis, lipogenesis, and fatty acid oxidation, directly or indirectly affecting host health [11]. The diversity of gut microbiota is another important factor associated with obesity. Many studies have shown that the species diversity and abundance of gut microbiota is reduced in obese patients [12,13]. In addition, gut microbiota is associated with abnormal lipid metabolism due to obesity.

The current interventions for obesity mainly include drug intervention, surgical treatment, and lifestyle changes; however, both drug intervention and surgical treatment have been shown to have certain undesirable side effects [3,4]. Among lifestyle change approaches, dietary interventions are more convenient, safe, and sustainable. Natural active ingredients such as dietary fiber [14], proteins [8], polysaccharides [15], polyphenols [16], and sterols [17] can prevent the effects of obesity caused by a high-fat diet (HFD). The plants containing these bioactive components have various beneficial functions, such as antioxidation and anti-inflammation; thus, they may be an ideal and efficient way to treat obesity and related metabolic diseases [5,8]. Bamboo shoots (BS) are the young shoots of *Qiongzhuea tumidinoda*, one of the rare species of bamboo endemic to China. According to the Chinese Pharmacopoeia, BS can cure thirst, relieve diaphragmatic discharges, reduce heat and phlegm (i.e., relief of fever, cough, and phlegm), and cool the stomach. BS is rich in active substances such as proteins, dietary fiber, polysaccharides, polyphenols, and sterols while being low in fat and soluble sugars, all of which have the potential to prevent obesity [18]. The current research has mainly focused on single components or extracts of BS. For example, sterol extracts [19], polysaccharides [20], and dietary fiber [21] from BS have been reported to improve prostatitis, lower blood sugar, and improve obesity. However, there are few studies on the use of BS powder to prevent obesity, and compared with oral drugs or probiotics [6,7], the intake of BS, a natural vegetable without side effects, is more in line with the daily diet of the human body. At the same time, the role of the gut microbiota and its metabolites in the improvement of metabolic dysfunction by BS has not been elucidated.

In the current study, we investigated whether BS exerted anti-obesity effects via the gut microbiota. Our study revealed that the underlying mechanism of BS alleviating obesity and its complications was linked to the modulation of gut microbiota and fecal microbiota metabolites.

## 2. Materials and Methods

### 2.1. Materials

Qiong BS (the young shoots of *Qiongzhuea tumidinoda*), provided by Shanyibao Biotechnology Co., Ltd. (Yiliang, China), was washed with clean water, cut into small pieces, frozen for 3 h in the −80 °C refrigerator (DW-HL398, Zhongke Meiling Cryogenic Technology Co., Ltd., Xiayi, China), and then transferred to the vacuum freeze dryer (LGJ-25C, Beijing Sihuan Scientific Instrument Factory Co., Ltd., Beijing, China) for drying. After drying the BS for 24 h, it was ground into powder with a high-speed pulverizer (WG-20B, Jinan Xinlu Biotechnology Co., Ltd., Jinan, China); next, finally, it was passed through an 80 mesh sieve, sealed, and stored for standby use. The basic nutritional components of freeze-dried BS powder were analyzed before the experiment (Appendix A). Other experimental reagent information is shown in Appendix A.

### 2.2. Animals and Diets

The animal care committee of the China Agricultural University (AW40601202-4-1) approved our animal experiments. After 1 week of adaptive feeding, five-week-old male C57BL/6J mice (Vital River Laboratory Animal Technology Co., Ltd., Beijing, China) were randomly divided into 4 groups (*n* = 8 in each group), each receiving a specific diet for 12 weeks: the NCD group (normal diet), the NCD-BS group, the HFD group (high-fat diet), and the HFD-BS group. The diets of mice in the NCD-BS and HFD-BS groups were supplemented with 15% BS lyophilized powder. The recommended daily intake of vegetables is 300–500 g for adults, according to the “Dietary Guidelines for Chinese Residents 2021”, and 15% BS was added based on previous studies [21]. Body weight and food intake were measured weekly. At the end of the experiment, blood was collected from the eyeballs after collecting feces. Mice were dissected, and epididymal fat, perirenal fat, groin fat, and liver tissue were collected for subsequent experiments. The “Shuyishuer Biotech Co.” provided the food required for the study. We demonstrate the specific composition and caloric content of each diet in Appendix A.

### 2.3. Histological Analysis

Adipose tissue was stained with hematoxylin and eosin (H&E), and liver tissue was stained with H&E and Oil-Red O. The tissue sections were then placed under a microscope, several randomly selected fields of view were photographed, and the size of the adipocytes was measured using Image J software. At the same time, the degree of steatosis of each mice liver was scored according to the evaluation criteria in Appendix A.

### 2.4. Determination of Blood Biochemical Parameters in Serum

The determination of lipopolysaccharide (LPS, limit of detection 1 EU/L) and tumor necrosis factor α (TNF-α, limit of detection 20 pg/mL) was performed using an enzyme-linked immunosorbent assay (ELISA) kit. A fully automatic biochemical analyzer (AU480, Olympus Corporation, Tokyo, Japan) was used to determine total serum triglyceride (TG), total cholesterol (TC), low-density lipoprotein cholesterol (LDL-C), and high-density lipoprotein cholesterol (HDL-C). 

### 2.5. Glucose Tolerance Test

During the 11th week of the experiment, mice were fasted for 12 h. A glucose tolerance test (ipGTT) was performed by injecting the mice intraperitoneally with 1.0 g/kg of glucose dilution solution. The glucose concentration in the tail venous blood was detected by a glucose meter (AccU-Chek, Roche, Switzerland).

### 2.6. Determination of Insulin in Serum

Insulin was measured using a mouse insulin enzyme-linked immunosorbent assay (ELISA) (Alpco, Salem, NJ, USA), and a homeostasis model assessment of insulin resistance (HOMA-IR) was calculated.

### 2.7. Gut Microbiota Analysis

This part of the experiment mainly includes three processes: the extraction of genomic DNA of fecal microorganisms, the amplification and sequencing of *16S rRNA* gene of microorganisms, and the processing and analysis of sequencing data. The specific methods for each of these processes are described in Appendix A.

### 2.8. Quantification of Fecal Short-Chain Fatty Acids (SCFAs)

The SCFAs were analyzed by gas chromatography (GC) spectrometry, following the protocol of Hu et al. [22]. Details of the specific method are shown in Appendix A.

### 2.9. Determination of Metabolites in Feces

The metabolites in feces were analyzed by LC-MS/MS. Details of the specific method are shown in Appendix A.

### 2.10. Statistical Analysis

Statistical analyses were performed with SPSS 25.0 (SPSS Inc., Chicago, IL, USA). All data are shown as mean ± SEM (standard error of mean). One-way ANOVA, followed by Duncan’s test (*p* < 0.05), determined significant differences in different groups. Graphs were prepared by GraphPad Prism 7.0 (La Jolla, CA, USA). The raw 16 s rRNA sequencing data have been deposited in the NCBI Sequence Read Archive (SRA) (http://www.ncbi.nlm.nih.gov/sra/ (accessed on 1 July 2022) under the Bioproject ID PRJNA915634.

## 3. Results

### 3.1. BS Alleviated Obesity Induced by HFD

Starting at the second week, the body weights of the mice in the HFD group were significantly higher than those of the mice in the NCD group. This trend of increasing body weight became obvious over time, indicating that our mice obesity model had successfully been established. As shown in Figure 1A, BS supplementation significantly reduced HFD-induced weight gain from the third week onward. The epididymal, groin, and perirenal fat weights were reduced in the HFD-BS group compared to the HFD group, while there was no significant difference in the weights of different fat types between the NCD and NCD-BS groups (Figure 1B). These results were produced without affecting the energy intake of mice (Appendix A). Indicators of lipid metabolism in the serum of mice were further measured. As shown in Figure 1C, the levels of serum lipid indices, such as TC, TG, HDL-C, and LDL-C, were significantly reduced in the HFD-BS group compared with the HFD group, indicating that BS might improve the abnormal lipid metabolism caused by HFD. At the same time, we found that the levels of serum lipid indices in mice of the NCD-BS group were not statistically different from those of the NCD group (Figure 1C). The HE staining and cell area calculation results of the epididymal fat further explained the changes in cell size and morphology of fat between different groups, and both showed that BS had significantly reduced the area of white fat cells in mice fed a high-fat diet (Figure 1D and Appendix A).

In general, obesity is accompanied by lipid deposition and inflammatory infiltration in the liver. The results of HE staining in mouse livers showed that the hepatocytes of mice on a normal diet, with or without BS intervention, had normal morphologies and no lipids. In contrast, HFD-fed mice had significantly enlarged and lipid-filled liver cells, as well as the formation of fat cysts with higher histological scores, but both of these effects improved after BS supplementation (Figure 2A and Appendix A). The morphology of hepatocytes in the HFD-BS group was similar to that in the NCD group, and no obvious fat cysts were found in either group (Figure 2A). The results of Oil-Red O staining in mouse livers supported the results of the HE staining. The high-fat diet led to the formation of many fat droplets in the liver, but the white fat droplets were significantly reduced after BS intervention (Figure 2B). The release of the proinflammatory factor TNF-α is associated with cellular immune responses, and it is induced by lipopolysaccharide (LPS), a major component of the cell walls of Gram-negative bacteria. There was no significant difference in serum LPS and TNF-α levels between the NCD group and the NCD-BS group, indicating that consumption of BS does not cause inflammation. Instead, the levels of LPS and TNF-α were significantly elevated in HFD mice, indicating that the obese mice had certain degrees of inflammation. After BS supplementation, the two inflammatory factors were significantly decreased, suggesting that BS had an anti-inflammatory effect (Figure 2C,D).

Dysregulated lipid metabolism is closely related to glucose metabolism balance and insulin signal transmission. In this sense, it is necessary to measure blood glucose and insulin-related indicators in mice. We can see that in the HFD group, the fasting blood glucose reached 7.57 ± 0.50 mmol/L and insulin reached 2.12 ± 0.60 ng/mL, both significantly higher than in the NCD group. After BS intervention, fasting blood glucose decreased to 6.27 ± 0.52 mmol/L and fasting insulin decreased to 1.02 ± 0.21 ng/mL, both significantly lower than in the HFD group. Fasting blood glucose and insulin were 3.61 ± 0.16, 4.31 ± 0.23 mmol/L, 0.47 ± 0.05, and 0.52 ± 0.07 ng/mL in the NCD-BS group and the NCD group, respectively. There were no significant differences between the two groups (Figure 3A,B). IpGTT results showed that the blood glucose levels of the HFD group were 27.01 ± 1.54 mmol/L 15 min after the intraperitoneal injection of glucose, 30.69 ± 0.84 mmol/L after 30 min, 24.31 ± 1.27 mmol/L after 60 min, 15.90 ± 1.30 mmol/L after 90 min, and 10.78 ± 0.63 mmol/L after 120 min. After BS intervention, they were significantly reduced to 21.47 ± 2.43, 25.67 ± 2.57, 15.76 ± 1.74, 12.60 ± 1.34, and 8.30 ± 0.62 mmol/L, respectively (Figure 3C). The HOMA-IR and area under the curve (AUC) calculated for blood glucose and insulin showed similar results (Figure 3D,E). Our study confirmed that BS intervention can increase insulin sensitivity and improve the hyperglycemia caused by HFD.

### 3.2. BS Improved Gut Dysbiosis in Obese Mice

The gut microbiota is closely associated with obesity, and dysregulation of the gut microbial community affects many metabolic functions. There are three main spatial scales for the determination of microbiota biodiversity: alpha diversity, beta diversity, and gamma diversity. Alpha diversity analysis reflects the richness, evenness, and diversity of microbial communities. The abundance-based coverage estimator (ACE) and the Chao1 estimator (Chao) indices show community richness, while the Heip and Shannoneven indices show community evenness. As shown in Figure 4A,B, BS supplementation increased the community richness that had been decreased by HFD, and the NCD-BS group also showed greater bacterial richness (Figure 4A,B). There were no significant differences in the Heip and Shannoneven indices between the four groups (Figure 4C,D), indicating that BS supplementation did not affect the uniformity of the microbiota. However, the gut microbiota of the four different treatment groups clustered relatively separately, implying that BS can alter the gut microbiota composition of both normal weight and obese mice (Figure 4E,F).

At the phylum level, Firmicutes, Bacteroidota, Proteobacteria, Desulfobacterota, and Verrucomicrobiota are the main microbiota (Figure 4G). In order to further compare the differences between the different groups at each phylum level, we conducted further statistical analyses. We found that, compared with HFD group mice, in the HFD-BS group, Bacteroidota was significantly increased, and Firmicutes, Proteobacteria, and Desulfobacterota were significantly decreased (Appendix A). At the genus level, the abundance of *Lachnospiraceae_NK4A136_group*, *Norank_f_Muribaculaceae,* and *Akkermansia* was significantly reduced in the HFD group compared to the NCD group, whereas the concentration of *Blautia*, *Anaerotruncus*, and *Streptococcus* increased (Figure 4H). BS obviously reversed this trend. Intriguingly, the Wilcoxon rank-sum test showed that BS administration increased *Prevotellaceae_UCG-001*, *Eubacterium_xylanophilum_group*, *Eubacterium_siraeum_group*, *Clostridium_sensu_stricto_1*, *Ruminococcus*, and *Burkholderia–Caballeronia–Paraburkholderia* (Figure 4I). According to the cladogram corresponding to the phylogenetic levels (from phylum to genus) obtained from LEfSe analysis, specific changes in food interventions occurred due to specific changes in barriers (Appendix A). Furthermore, the LDA scores from LEfSe analysis indicated that the gut microbiota of HFD mice was richer in *p_Firmicutes*, *g_Blautia*, *g_unclassified_f_Lachnospiraceae*, *g_Anaerotruncus*, and *g_Lachnoclostridium* compared with the NCD mice. Adding BS not only reduced the growth of these bacteria, but also significantly increased the concentrations of *f_Muribaculaceae*, *g_Lachnospiraceae_NK4A136_group*, *g_Eubacterium_xylanophilum_group, f_Prevotellaceae*, *g_Romboutsia*, *g_Clostridium_sensu_stricto_1*, *g_Ruminococcus*, *g_Eubacterium_siraeum_group*, and *g_Prevotellaceae_UCG-001* (Appendix A).

### 3.3. BS Promotes the Production of SCFAs

The levels of acetate, propionate, butyrate, isobutyrate, valerate, and isovalerate in mice feces were significantly reduced by HFD. After BS intervention, the SCFA content was significantly increased, and the contents of acetate, propionate, butyrate, isobutyrate, valerate, and isovalerate were increased to 0.32 ± 0.02, 0.25 ± 0.02, 0.27 ± 0.03, 0.03 ± 0.01, 0.03 ± 0.01, and 0.05 ± 0.01 mg/g, respectively. The normal diet supplemented with BS mainly increased the acetate content to 0.30 ± 0.02 mg/g, compared with 0.22 ± 0.02 mg/g, in the NCD group (Figure 5).

### 3.4. Effects of BS on Fecal Metabolites in High-Fat Diet-Fed Mice

In positive and negative ion mode, the whole metabolism of mice in the NCD, HFD, and HFD-BS groups was predicted by orthogonal partial least squares (QPLS-DA). The results showed that the three groups were significantly separated (Figure 6A,B). The greater the degree of separation between the different treatment groups, the more significant the classification effect. BS was further separated from the HFD group after supplementation, indicating that the metabolic improvement effect was obvious (Figure 6A,B). Levels of estradiol, N’ -formylkynurenine, cortisol, (−) −stercobilin, and I-urobilin were significantly increased in the HFD group compared with the NCD group. BS intake ameliorated these changes, and also increased the contents of 10-nitrolinoleic acid, dihydroartemisinin (DHA), glutathione, 7-sulfocholic acid, adenosine diphosphate ribose, malic acid, N1, N12-diacetylspermine, L-aspartic acid, 9/10/13-TriHOME, undecenoic acid, L-Dopa, and other metabolites (Figure 6C). Annotation into the Kyoto Encyclopedia of Genes and Genomes (KEGG) functional pathway showed that these metabolites were mainly involved in the biosynthesis of lipids, amino acids, and other secondary metabolites (Figure 6D). These results suggest that BS metabolites may play an essential role in reducing lipids by regulating these obesity-related metabolic pathways.

## 4. Discussion

Obesity is a complex metabolic disease, and is prone to insulin resistance, white fat and liver lipid accumulation, inflammation, and abnormal lipid metabolism. So far, there are no safe and effective drugs that can effectively treat obesity; however, individual components or extracts from BS are effective in this sense [21,23]. In this study, we evaluated whether BS whole powder has similar anti-obesity effects as the single component which was previously studied. The use of BS powder to alleviate the challenges posed by obesity may represent a new method that is more relevant to realistic consumption habits. Our data demonstrate that the BS powder was effective in reducing body weight gain, lipid accumulation, insulin resistance, and inflammatory response in HFD-induced obese mice. Importantly, BS had significant effects on the gut microbiota and its metabolites. This modulatory effect on the gut microbiota and its metabolites may represent a mechanistic explanation of the positive effect of BS with regard to obesity.

The present study first investigated the effect of BS on obesity indicators in mice, namely, body weight, fasting glucose, lipid metabolism, and inflammatory response. Previous studies suggested that high levels of dietary fiber in BS powder may have a role in improving obesity. Consistently with these findings, we demonstrate the positive effects of BS on obesity status [21,23].

The gut microbiota is closely linked to obesity, and the interaction between dietary components and gut microorganisms determines the composition of the gut microbiota, thus having a significant impact on host metabolism. Analysis of the comparative *16S rRNA* gene sequencing results revealed that at the phylum level, the HFD-BS group significantly increased the abundance of Bacteroidota and decreased the abundance of Firmicutes. The studies have shown that Firmicutes can absorb energy more efficiently than Bacteroidota, thus promoting energy absorption and body weight gain [24,25]. At the genus level, the abundances of *Lachnospiraceae_NK4A136_group* and *Norank_f_Muribaculaceae* were increased in the HFD-BS group, while those of *Blautia* and *Burkholderia–Paraburkholderia* were decreased. The *Lachnospiraceae_NK4A136_group* showed beneficial anti-inflammatory effects by reducing lipid levels in the liver and serum [26,27]. The high abundance of *Norank_f_Muribaculaceae* is beneficial for reducing serum levels of TG, TC, LPS, and TNF-α, as well as improving obesity in general [28,29]. *Blautia* was reported to be positively correlated with obesity and glucose tolerance [30]. The *Burkholderia–Paraburkholderia* genus contains several pathogenic strains with pro-inflammatory properties [31]. Thus, *Blautia* and *Burkholderia–Paraburkholderia* may be detrimental to improving obesity. These findings show that the anti-obesity effect of BS is exerted via regulation of the gut microbiota.

The different effects of BS powder and BS dietary fiber on gut microbiota are also noteworthy. This observation suggests that other components of BS powder aside from dietary fiber may be involved in the regulation of gut microbiota, and that the joint regulation of multiple components produces different results. Further, because of the significant differences in nutrient content between different types of bamboo shoots, we suggest that bamboo shoot species may also influence the gut microbiota, albeit differently. The most commonly used bamboo shoots in existing studies are the Mao, Lei, Fang, Ma, and Banna sweet dragon bamboo shoots, which are high in dietary fiber [32], whereas the Qiong bamboo shoots used in this study contain more protein [33]. More in-depth experimental studies are needed to analyze the effects of different components of bamboo shoots on obesity improvement, in addition to the inter-regulatory effects between the BS powder components and gut microbiota.

Compared to the HFD group, the HFD-BS group not only increased the levels of beneficial bacteria that produce short-chain fatty acids (SCFAs), but also decreased the levels of harmful bacteria. These results provide a good basis for improving obesity by modulating the gut microbiota. Among the beneficial bacteria is *Lachnospiraceae NK4A136,* which can produce acetate and butyrate by breaking down dietary fiber in BS [34,35]. *Norank_f_Muribaculaceae* is a potential propionate-producing probiotic [36]. In addition, in the HFD−BS group, the levels of bacteria known to promote acetate, propionate, and butyrate production were enriched, including *PrevotelaceAE_ UCG-001*, *Eubacterium Xylanophilum Group*, *Eubacterium_siraeum_group*, and *Clostridium sensu stricto* [37,38,39,40,41]. SCFAs are important gut microbiota metabolites that maintain intestinal physiological functions; they have beneficial effects on insulin sensitivity and a role in the mitigation of metabolic diseases such as obesity and diabetes [37]. Acetate and propionate can inhibit lipid synthesis in the liver, and butyrate can lower blood glucose [8]. This may be the reason why increasing levels of acetate and propionate in the HFD-BS group can improve obesity.

In this study, a non-targeted metabolomic analysis of mice feces was performed. The metabolites with high levels that differed significantly between groups, and which were also highly correlated with the development of obesity, are listed in Figure 6C. These metabolites are mainly fatty acids and their metabolites (10-nitrolinoleic acid, 9,10,13-TriHOME, ricinoleic acid, and undecylenic acid) and bile acids (7-sulfocholic acid), which are closely related to lipid metabolism. 10-nitrolinoleic ac-id, undecylenic acid, and L-aspartic acid all have strong anti-inflammatory and antioxidant effects, and they help to reduce serum inflammatory factor levels [42,43,44]. Interestingly, *Lachnospiraceae_NK4A136_group* and *Norank_f_Muribaculaceae*, which were significantly increased in the HFD-BS group, were also able to exert beneficial anti-inflammatory effects by lowering liver and serum lipid levels [26,27,28,29]. Therefore, BS may have further enriched associated fatty acid metabolites by increasing beneficial gut microbiota, thereby reducing inflammation levels and improving obesity lipid metabolism. We also found that improved lipid metabolism in mice after BS intervention was achieved by fatty acid-like and bile acid-like metabolites through activation of PPAR. Peroxisome proliferator-activated receptors (PPARs) are fatty acid sensors that regulate the whole-body metabolism [45]. Of these, 10-nitrolinoleic acid is an endogenous lipid mediator and agonist of PPAR [46], and 7-sulfocholic acid, lithic cholic acid (LCA) [47], and 9,10,13-Trihydroxy-11-Octadecenoic Acid (9,10,13–TriHOME) act as linoleic acid-derived lipid mediators. Both of these, in combination with linoleic acid, can also activate PPAR [48]. In summary, BS can increase the beneficial bacteria in *Lachnospiraceae_NK4A136_group* and *Norank_f_Muribaculaceae* and enrich fatty acid-like metabolites, thereby reducing lipid levels in vivo, activating the PPAR signaling pathway, and regulating lipid metabolism in mice, thus improving obesity.

Finally, compared with previous studies using siglinine-like drugs [7], probiotics [6], or grain extracts [8] to prevent obesity and related metabolic diseases, we innovatively demonstrated that BS, a natural vegetable-like substance, showed a significant effect in preventing obesity. Compared to drugs and grain extracts [7,8], BS is safer, more convenient, and more sustainable for the prevention of obesity, and, more importantly, it is in line with typical human daily eating habits. At the same time, the mechanism of BS in alleviating obesity may be to activate the PPAR signaling pathway by enriching beneficial gut microbiota, which then produces SCFAs and increases fatty acid metabolites. This is different from the mechanism of *Bifidobacterium* [6], which prevents obesity by increasing mitochondrial bioproduction and adipose tissue function.

## 5. Conclusions

This study provides clear evidence that BS can improve obesity caused by HFD. BS can significantly improve obesity-related phenotypes, such as liver steatosis, insulin resistance, and inflammatory reactions. It can also regulate the composition and structure of gut microbiota and promote the enrichment of fatty acid metabolites. We propose that BS may improve obesity by reshaping gut microbiota and metabolic function. Thus, BS has great potential as a functional food for preventing and treating obesity. Our study provides a theoretical basis for the future utilization of bamboo shoot resources as effective components in improving obesity.

## Figures and Tables

**Figure 1 foods-12-01380-f001:**
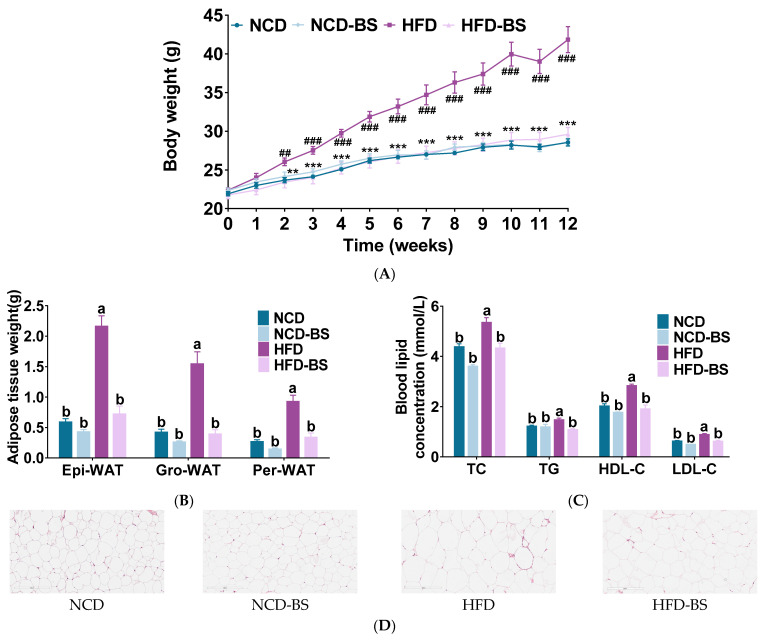
BS alleviated obesity induced by HFD. (**A**) Body weight time course measurements, (**B**) white fat weight, (**C**) serum lipid profile, and (**D**) H&E staining of epididymal fat sections. Data are presented as mean ± SEM, *n* = 8 per group. ^##^ *p* < 0.01, ^###^ *p* < 0.001, HFD versus NCD; ** *p* < 0.01, *** *p* < 0.001, HFD-BS versus HFD. Means are denoted by a different letter (a, b) indicate significant differences between groups (*p* < 0.05). Epi-WAT indicates epididymal fat; Per-WAT, perirenal fat; Gro-WAT, groin fat; NCD, normal control diet; NCD-BS, normal control diet supplemented with freeze-dried powder of bamboo shoots; HFD, high-fat diet; HFD-BS, high-fat diet supplemented with freeze-dried powder of bamboo shoots.

**Figure 2 foods-12-01380-f002:**
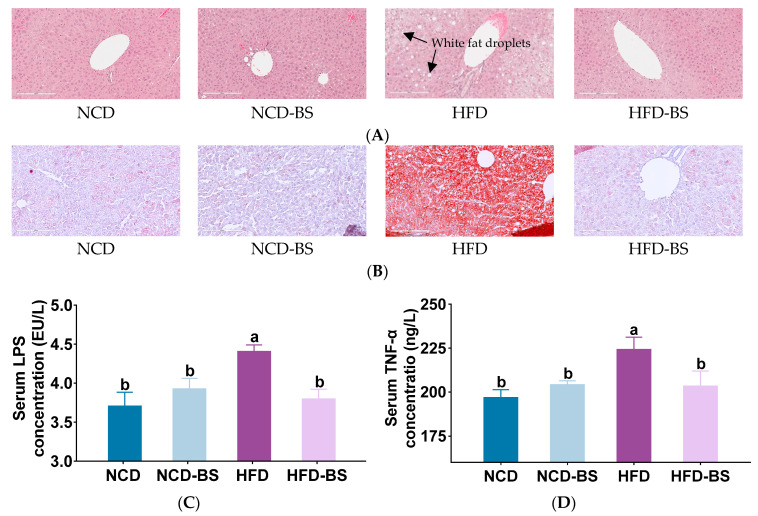
BS limited HFD-induced hepatic steatosis and ameliorated inflammation. (**A**) H&E staining of liver tissue, (**B**) Oil-Red O staining of liver tissue, (**C**) serum LPS concentrations, and (**D**) serum TNF-α concentrations. Data are presented as means ± SEM, *n* = 8 per group. Means denoted by a different letters (a, b) indicate significant differences between groups (*p* < 0.05). NCD, normal control diet; NCD-BS, normal control diet supplemented with freeze-dried powder of bamboo shoots; HFD, high-fat diet; HFD-BS, high-fat diet supplemented with freeze-dried powder of bamboo shoots.

**Figure 3 foods-12-01380-f003:**
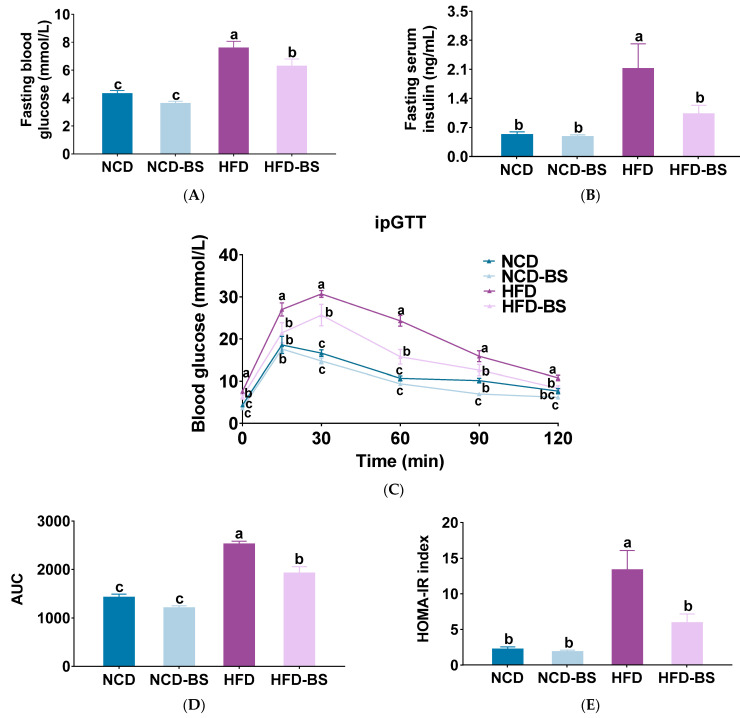
BS improved insulin resistance and glucose tolerance in HFD-fed mice. (**A**) Serum glucose, (**B**) serum insulin, and (**C**) blood glucose levels over time during the intraperitoneal glucose tolerance test (iPGTT); (**D**) homeo-static model assessment for insulin resistance (HOMA-IR) index, and (**E**) area under the curve calculation (AUC). Data are presented as mean ± SEM, *n* = 7. Means denoted by different letters (a, b, c) indicate significant differences between groups (*p* < 0.05). NCD, normal control diet; NCD-BS, normal control diet supplemented with freeze-dried powder of bamboo shoots; HFD, high-fat diet; HFD-BS, high-fat diet supplemented with freeze-dried powder of bamboo shoots.

**Figure 4 foods-12-01380-f004:**
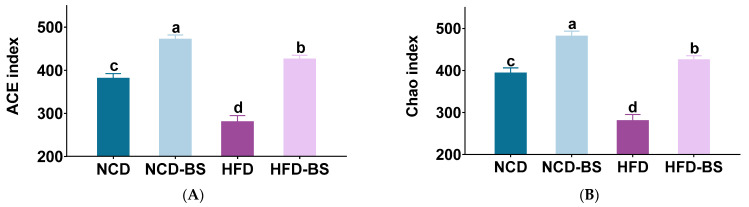
BS attenuated gut microbiota dysbiosis in HFD−fed mice. (**A**,**B**) Community richness, accessed by the ACE and Chao indices. (**C**,**D**) Community diversity, accessed by the Heip and Shannoneven indices. Means denoted by a different letter (a, b, c, d) indicate significant differences between groups (*p* < 0.05). (**E**,**F**) PCoA and NMDS score plots based on Bray−Curtis. (**G**) The abundances of gut microbiota at the phylum level. (**H**,**I**) Mean proportions of key genera in different groups at the genus level based on the Wilcoxon rank-sum test. * *p* < 0.05, ** *p* < 0.01, *** *p* < 0.001 (in **H**,**I**). *n* = 8 per group. NCD, normal control diet; NCD−BS, normal control diet supplemented with freeze−dried powder of bamboo shoots; HFD, high−fat diet; HFD−BS, high-fat diet supplemented with freeze-dried powder of bamboo shoots.

**Figure 5 foods-12-01380-f005:**
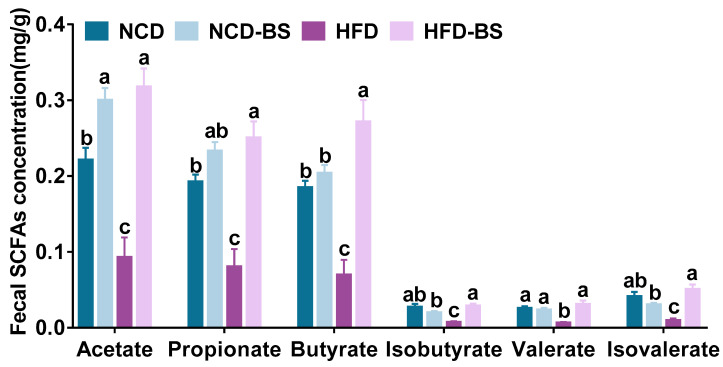
BS promoted the production of SCFAs. *n* = 6 per group; data are shown as means ± SEM. Means denoted by different letters (a, b, c) indicate significant differences between groups (*p* < 0.05). NCD, normal control diet; NCD−BS, normal control diet supplemented with freeze−dried powder of bamboo shoots; HFD, high−fat diet; HFD−BS, high−fat diet supplemented with freeze−dried powder of bamboo shoots.

**Figure 6 foods-12-01380-f006:**
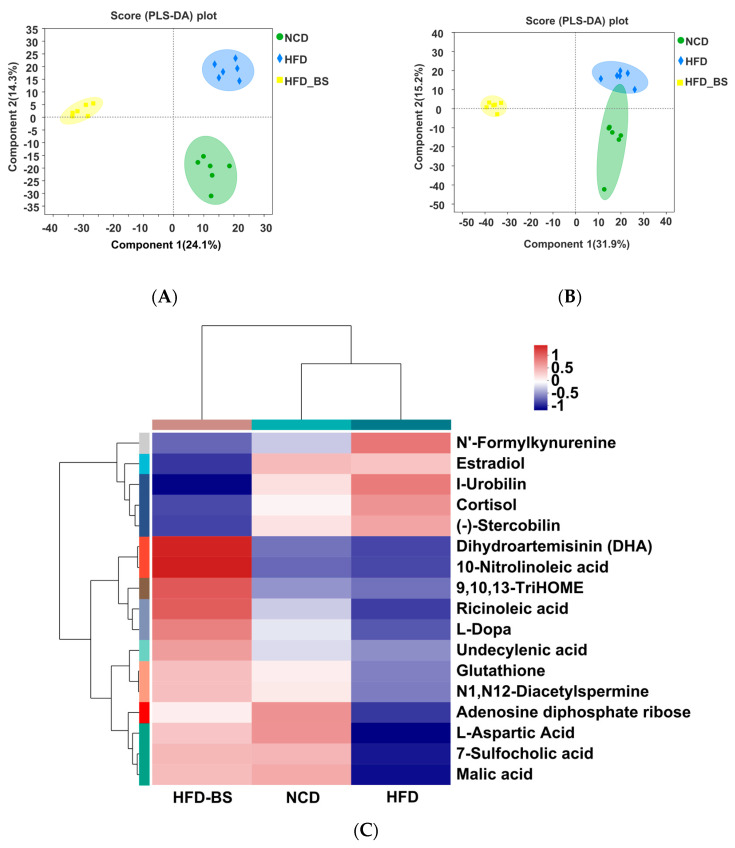
BS reversed the effects of the HFD on the serum metabolomic profiles of mice. Partial least squares analysis (PLS−DA) scores of the fecal metabolomics of the three groups are shown. (**A**) Positive ion; and (**B**) negative ion. (**C**) Hierarchical cluster analyses of differential metabolites. Red and blue colors with increasing intensity indicate up-regulation or down-regulation, respectively. The metabolites with VIP > 1 and *p* < 0.05 were determined as statistically significantly different metabolites based on the variable importance in the projection (VIP) obtained by the OPLS−DA model and the *p*-value generated by Student’s *t* test. (**D**) KEGG functional annotation pathway map. *n* = 6 per group, KEGG, Kyoto Encyclopedia of Genes and Genomes; NCD, normal control diet; NCD−BS, normal control diet supplemented with freeze−dried powder of bamboo shoots; HFD, high-fat diet; HFD−BS, high−fat diet supplemented with freeze−dried powder of bamboo shoots.

## Data Availability

Data is contained within the article or Appendix A.

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
