# Peer review of "Bamboo Shoots Modulate Gut Microbiota, Eliminate Obesity in High-Fat-Diet-Fed Mice and Improve Lipid Metabolism"

_foods, 2023, doi:10.3390/foods12071380_

Round 1

Reviewer 1 Report

This study examines the effect of bamboo shoots (BS) on a high fat diet in an obesity mouse model. The authors find that adding 15% BS (in a freeze-dried powder form) largely mitigates the effect of a high fat diet in terms of weight gain and several other metabolic and physiologic measures. The gut microbiome of the mice fed BS was altered compared to both the control and experimental diet. This study adds to the literature examining the role of various food supplements that may have beneficial effects on physiology and metabolism.

The work presented here is largely sound. I found no significant fault with the study design or the conclusions. The effects seen were significant and clear. The conclusions were supported by the data. That being said, I would like to see some explanation of the diets. There were several differences among the diets other than the presence/absence of BS powder and lard added to make the high fat diet. For example, the high fat diet had over 13X as much vitamin mix compared to the control diet. The high fat diets also contained more maltodextrin and casein, among other differences. I have no expertise in rodent diets, so I readily admit this may be standard practice and well documented, but I would like to see at least a reference to where the diets came from or something about how the diets were designed. Without that, I can’t be sure that the differences observed are due only to the excess fat in the high fat diet or the presence/absence of BS and not some other difference in the recipes of the diets. 

Figures S1 and S2 are missing. These can not be evaluated.

The manuscript has many instances of misspellings and awkward grammar, and a few instances of phrases that seem to be poorly translated from Chinese and make little sense in English. Careful editing should be done to correct the language.

The text in most of the figures (labels, etc.) is very small and difficult to read. These figures should be recreated with larger text.

In Figure 6D, the figure legend and the figure don’t seem to match. The legend mentions abbreviations that are not in the figure. The legend states it is a KEGG pathway enrichment map, but it is a histogram of pathway terms.

References 16 and 17 appear to be the same reference.

Reviewer 2 Report

The manuscript by Xiaolu Zhou et al. Bamboo shoots modulate gut microbiota, eliminate obesity in high-fat diet-fed mice, and improve lipid metabolism” demonstrated that bamboo shoots (BS) supplementation has a great potential as a functional food for preventing and treating obesity. Using an experimental high-fat diet (HFD) model in mice, the authors showed that BS powder was effective in reducing body weight gain, lipid accumulation, insulin resistance, and inflammatory response in murine obesity. Additionally, BS was able to regulate the gut microbiota, resulting in increased levels of beneficial bacteria that produce short-chain fatty acids (SCFAs), and decreased levels of harmful bacteria. Finally, BS had a significant modulating effect on fecal metabolites, such as fatty acids and bile acids, which are highly relevant to the altered gut microbiota. Based on the modulatory effect of BS supplementation on microbiota composition and gut metabolite levels, the authors suggest that BS may be beneficial in obesity and its related complications.

Together, this is an interesting manuscript, which was full well conducted and presents original data concerning the protective effect of BS on HFD-induced gut dysbiosis in mice.  

Some minor aspects are raised in the manuscript that should be addressed by the authors.

1 1. In the Materials and Methods section, subsection 2.3, the authors refer to measuring the size of adipocytes and evaluating the degree of liver steatosis. However, the results of these measurements were not presented, only the representative tissue photomicrographs.

2 2. Some linguistic mistakes should be corrected and the whole manuscript should be revised by native English speaker. 

Reviewer 3 Report

The manuscript titled: “Bamboo shoots modulate gut microbiota, eliminate obesity in high-fat diet-fed mice, and improve lipid metabolism” shows

The article presents high similarity with previously published research, particularly:

Huo et al. (2020). https://doi.org/10.3390/molecules25071490

Wang et al. (2022). https://doi.org/10.3389/fendo.2022.866189

Zhao et al. (2022). https://doi.org/10.1002/mnfr.202100907

Authors are encouraged to reduce the similarity level of their manuscript. Major comments are suggested for this manuscript.

Abstract

1.     Line 16: Please write 16S rRNA in italics as this is a gene.

2.     Line 18: Numeric data are needed. For instance: how much were the body weight reduced? (Percentage or values).

3.     Line 19: Please correct: short-chain fatty acids (SCFAs). This should be corrected along with the manuscript.

4.     Line 22: What do “modulating” means? Is it increasing or decreasing fecal metabolites?

Keywords

5.     “Metabolic” seems incomplete. Please find a suitable keyword. Moreover, please arrange keywords in an alphabetical manner.

Introduction

6.     Lines 30-32: Obesity is a multifactorial disease, so the sentence indicating that just the imbalance between energy intake and expenditure is not correct.

7.     Lines 33-34: What do the authors mean with “fat accumulation infiltrates immune cells”? Does fat affect the immune cells? Authors should elaborate more this sentence to be as accurate as possible.

8.     Lines 42-43: More than “decomposition”, it could be named as: carbohydrates bacterial metabolism, o similar.

9.     Lines 46-47: What are the undesirable effects of lifestyle changes?

10.  Line 47: Please correct: proteins (add an “s”).

11.  Line 48: “Improve obesity” could be misleading. I suggest the authors to change by: “improve metabolic outcomes of obesity” or “prevent obesity effects” or similar. This should be correct throughout the manuscript.

12.  Line 54: What do the authors mean with “reduce heat”? Please elaborate more this idea to give more information to the readers.

13.  Line 55: How do “trace elements” have the potential to improve obesity?

14.  More information regarding the overall impact of gut microbiota on the obesity condition is needed in the introduction.

Materials and methods

15.  Line 68: How did the authors prepared the freeze-dried powder? Information about the equipment used and the operational conditions are needed. If already published, authors should add a reference.

16.  Line 75: 5-weeks old mice are very young mice (early adult). Why did the authors choose this age? Most in vivo obesity experiments start using 7-10 weeks’ old mice for 12-16 weeks treatment.

17.  Lines 90-91: Please re-phrase.

18.  Line 95: Please add the detection limits of each ELISA test.

19.  Line 112: Please avoid starting sentences with “And”. Please find a suitable connector.

Results and discussion

20.  I don’t understand the way the letters were assigned in the statistical analysis. For instance, in Fig. 1 it seems that “b” denotes the highest values and “a” the lowest (and bars sharing the same letters are not significant). However, in Figure 3, “C” should be denoted for the tallest bar, but it is not in this case. Usually, “a” is the highest value (or the major values), followed by “b”, and so on. Please double check the statistical analysis.

21.  Line 137: Usually, the HFD contains more calories, and animals become satiated, consuming less food at the end of the experiment. How do the authors explain that no changes in the food intake of mice were achieved?

22.  Line 164: Please avoid the word “obvious”.

23.  Line 232: Please spell ACE. Is it Chao1?

24.  Line 233: Please correct: “Shannon index”.

25.  Line 390: Please correct: “non-targeted metabolomic”.

Round 2

Reviewer 3 Report

Dear authors, thank you for the effort made to improve the scientific quality of this manuscript. I believe it can be now accepted for publication in Foods. Best regards

Author Response

Many thanks to the reviewer for your acknowledgment of our manuscript revisions.
Thank you again for all your professional advice on our manuscript! Best regards.